# Comparison of radiation belts electron fluxes simultaneously measured with PROBA-V/EPT and RBSP/MagEIS instruments

Alexandre Winant[1,2], Viviane Pierrard[1,2], and Edith Botek[1]

[1]Royal Belgian Institute for Space Aeronomy (BIRA-IASB), Brussels, Belgium
[2]Université Catholique de Louvain, Earth and Life Institute ELI-C, Louvain-La-Neuve, Belgium

**Correspondence:** Alexandre Winant (alexandre.winant@aeronomie.be)

**Abstract.** Relativistic radiation belt electron observations from the Energetic Particle Telescope (EPT) onboard the PROBA-V satellite are compared to those performed by the Magnetic Electron Ion Spectrometer (MagEIS) onboard the Van Allen Probes formerly known as the Radiation Belt Storm Probes (RBSPs). Despite their very different orbits, both instruments are able to measure fluxes of electrons trapped on a given magnetic shell. In the outer belt, the comparison of high and low altitude fluxes is performed during the first three months of 2014, featuring the most intense storms of the year. In the inner belt, measurements from the two instruments are compared only at conjunction, when the satellites are physically close to each other. Due to the low number of conjunctions, the whole period of mutual operation of both instruments is used (i.e. May 2013-October 2019). The comparisons show that flux variations appear simultaneously on both spacecraft, but the fluxes observed by the EPT are almost always lower than for MagEIS, as expected from their different orbits. In addition, this difference in flux intensity increases with electron energy. During geomagnetic storms, it is also shown that dropout events (i.e. sudden depletion of electrons) in the outer belt are more pronounced at low altitudes than near geomagnetic equator. The effect of the equatorial pitch angle value of electrons is investigated in the outer belt. Despite the difference in flux intensity observed by the two instruments, especially at high energies, a linear relationship with a linear correlation higher than 0.7 was found. The correlation is maximum when low pitch angle electrons near the equator are considered.

## 1 Introduction

The radiation belts are two toroidal regions that surround the Earth and are filled with highly energetic charged particles trapped in its geomagnetic field. The belts are separated by a slot region with very low fluxes of particles during quiet conditions (Koskinen, 2022). In terms of the McIlwain (1961) parameter L , the inner belt, composed of both protons and electrons of high energy, extends up to L = 2, depending on the particles energy, and presents a more stable configuration (see e.g Pierrard et al. (2022a) for protons measured by the EPT). The outer radiation belt, mainly composed of electrons, is highly sensitive to the geomagnetic activity induced by the interaction between the solar wind and Earth's magnetosphere. The dynamics of the radiation belts is extremely complex. The radiation belts particles are constantly added from various sources and lost due to different physical processes. A full review of the radiation belts dynamics was conducted by Ripoll et al. (2020). Critical physical processes to consider in the radiation belts are the wave-particle interactions between cold plasma and the high energy

particles of the belts. The plasmasphere, a region of cold and dense plasma originating from the ionosphere (Goldstein, 2007), overlaps with the radiation belts. The different densities found inside and outside the plasmasphere generate different types of waves that can lead to particle losses in the belts. The power of the waves present in the plasmasphere increases with plasma density which also vary with geomagnetic activity. Thus variations of density directly influence the diffusion coefficients that characterize the wave-particle interactions in the radiation belts (Ripoll et al., 2023). During geomagnetic storms, electron fluxes can decrease and increase abruptly in a few hours (Pierrard and Lopez Rosson, 2016; Reeves et al., 2016), and cause numerous problems to satellite systems such as surface and internal charging. Due to the hazard posed by such populations, it is of prime importance to accurately measure and understand high energy electron fluxes.

Over the last decade, instruments entirely dedicated to the study of the radiation belts were developed and sent on diverse orbits around the Earth, such as the Magnetic Electron and Ion Spectrometer (MagEIS) (Blake et al., 2013) launched in 2012 onboard the Van Allen Probes on a highly elliptic equatorial orbit (Mauk et al., 2013), the Energetic Particle Telescope (EPT) launched in 2013 on the PROBA-V satellite on a low polar orbit (Dierckx et al., 2014) and more recently the High-energy Electron Experiments (HEP) (Mitani et al., 2018) and the extremely high-energy electron experiment (XEP) (Higashio et al., 2018) on the ARASE satellite launched in December 2016 also in an equatorial trajectory (Miyoshi et al., 2018). The Van Allen Probes, already decommissioned in 2019, led to numerous discoveries about the radiation belts, including the detection of a third ultra-relativistic electron belt (Baker et al., 2013) or the discovery of an impenetrable barrier to ultra-relativistic electrons in the inner belt (Baker et al., 2014), which was confirmed at low altitudes by EPT observations (Pierrard et al., 2019). The observations from the instruments on-board the Radiation Belt Storm Probes (RBSPs), which have extensively been validated, are thus used as a standard to compare with instruments on ARASE (Sandberg et al., 2021; Szabó-Roberts et al., 2021) and on the GOES-15 in geostationary orbit (Baker et al., 2019). In addition, recent studies have compared electron fluxes observed in the outer radiation belt at low and high latitudes. Ginisty et al. (2023a) have taken advantage of the Electric Orbit Raising (EOR) of CARMEN4 to geostationary orbit to compare simultaneous observations at LEO of CARMEN3. Both missions were developed by the Centre National d'Etudes Spatiales (CNES) and are fitted with the same instrument, the ICARE-NG detector (Boscher et al., 2014). In this study, a linear relationship between logarithmic values of the electron fluxes $\geq 1.6$ MeV at low and high altitude was found between $L^* = 3.5 - 4.8$, where $L^*$ is the Roederer parameter (Roederer and Lejosne, 2018). In Ginisty et al. (2023b) a similar comparison is undertaken between CARMEN2-3 at LEO on JASON2 and 3 satellites with an orbit very different from PROBA-V, at an altitude of 1336 km and 66° of inclination and RBSP in the outer belt for relativistic electrons ($\geq 1.6$ MeV). In this work, they report that flux levels are quite similar for both mission, with a good linear correlation between $L^* = 3.5 - 4.8$.

In the present paper, observations from the PROBA-V/EPT are compared to observations from RBSP/MagEIS in the inner and outer belts. As for the GOES-15 satellite (Baker et al., 2019), there are only few moments of conjunction between PROBA-V and RBSP due to their very different orbits (low Earth polar orbit versus highly elliptic equatorial orbit, respectively). Conjunction periods are optimal to compare and validate measurements from two satellites since they are physically close to each other and share the same radiative environment. In the case of the PROBA-V satellite, these conjunctions could only occur in the South Atlantic Anomaly (SAA), when the RBSPs are at their perigee, and thus in the inner belt. However, due

to the motion of trapped particles in the geomagnetic field, both the EPT and the MagEIS instrument can measure fluxes of electrons trapped on the same magnetic shells (Pierrard et al., 2021). A first statistical comparison between EPT and MagEIS measurements was conducted in the outer belt throughout June 2015 which featured an intense geomagnetic storm (Pierrard et al., 2022b). From this study, good alignment of the data from both instruments was found, but the analysis showed some important differences during the dropout event caused by the geomagnetic storm. Thus, a comparison of those two instruments allows to see the difference in fluxes observed in the outer belt at low altitudes and near geomagnetic equator. A description of both instruments used in this work is given in section 2, together with the used methodology. In section 3 the results are provided and discussed. First observations of the fluxes measured by the EPT throughout 2014 and a comparison of the EPT observations throughout February 2014 with the AE8 (Vette, 1991) empirical model of the radiation belts are presented. Then, results of the comparison with two types of data sets of MagEIS (level 2 spin averaged and level 3 pitch angle resolved data) are presented for fluxes in the outer belt and conjunctions for fluxes in the inner belt. Finally, the fourth section brings the conclusions of these correlation studies.

## 2   Instruments and Methodology

### 2.1   EPT

The Energetic Particle Telescope (EPT) measures fluxes of high energy particles in the radiation belts. This instrument was developed by the Center for Space Radiation (CSR) at UCLouvain in Belgium, with the collaboration of the Royal Belgian Institute for Space Aeronomy and QinetiQ Space. This instrument has been launched in 2013 onboard the ESA satellite PROBA-V. The spacecraft was sent to a sun-synchronous LEO polar orbit at an altitude of 820 km, with an orbit inclination of 98.73° and a descending node at 10:30 am local time (Pierrard et al., 2014). The concept of the EPT is based on the Bethe-Block formula giving the relationship between the stopping power of a material and the energy of incident charged particles, this instrument is a so called $\Delta E - E$ telescope (Cyamukungu and Grégoire, 2011). The EPT was designed for real-time and contamination-free measurements of charged particle spectra in the space environment and is able to discriminate between electrons, protons, alpha particles and heavier ions while performing direct measurements of their energy spectra (Cyamukungu et al., 2014). The EPT features two energy sections. The Low Energy Section (LES) only measures lower energy electron fluxes, while the High Energy Section (HES) measures fluxes of higher energy electrons, protons and heavier particles. The EPT allows to measure flux of electrons above 500 keV in 6 energy channels, and protons above 9.5 MeV in 10 energy channels. The EPT data are available on https://swe.ssa.esa.int/space-radiation.

### 2.2   MagEIS

The Magnetic Electron Ion Spectrometer (MagEIS) is a science class spectrometer whose purpose is to measure fluxes of particles in the radiation belts. Unlike the EPT, MagEIS relies on uniform magnetic fields to focus electrons and sort their energy on a linear strip of detectors (Blake et al., 2013). This instrument is part of a larger suite of instruments specifically

**Table 1.** EPT and MagEIS channels compared in this work

| EPT | MagEIS |
| --- | --- |
| 500-600 keV | 558-639 keV |
| 700-800 keV | 692-793 keV |
| 800-1000 kEV | 840-952 keV |
| 1000-2400 keV | 970-1279 keV |
| 2400-8000 keV | 2280-3008 keV |

designed to study the radiation belts that was carried by the NASA satellites, Radiation Belt Storm Probes (RBSPs) (Boyd et al., 2019). The RBSP spacecraft were twin satellites, RBSP-A and RBSP-B, launched in 2012 on Geostationary Transfer Orbit (GTO) near the geomagnetic equator, with an orbit inclination of 10 degrees. This orbit is very elliptic so that at the apogee, the RBSP were near geostationary orbit (L $\sim$ 6.6), while the altitude of the perigee is around 600 km. The MagEIS instrument is composed of four magnetic spectrometers that measure fluxes in four energy ranges. MagEIS features a low energy unit (20-240 keV), two medium energy units (80-1200 keV) and a high energy unit (800-4800 keV) (Claudepierre et al., 2015). Those combined units give a wide energy range for the measured electron fluxes (20 keV-4 MeV) on a larger number of channels than for the EPT. MagEIS level 2 and level 3 data were retrieved from https://rbsp-ect.newmexicoconsortium.org/data_pub/ and only the background corrected MagEIS electron fluxes have been used all along the present work. Level 2 data are the spin-averaged (averaged on the spin of the spacecraft) fluxes measured by the instrument, while level 3 data provide fluxes of electrons in given pitch angle bins.

### 2.3 Methodology

Both instruments (EPT and MagEIS) measure the differential fluxes of particle (given in $s^{-1}cm^2sr^{-1}MeV^{-1}$) in the radiation belts. However, some differences between them are important for the following comparison. First of all, the number of energy bins and their width are not the same. For electrons, the EPT has 6 usable energy channels ranging from 500 keV to 8000 keV, while MagEIS has 21 channels ranging from 33 keV to 4000 keV. Because the flux decreases with energy, in order to perform a meaningful comparison between the two instruments, the lower energy edge of the channels to be compared must be as close as possible. The channels that were compared in this work are shown in Table 1. Note that the second channel of the EPT (600-700 keV) was not used since there were no similar channel for the MagEIS instrument.

In addition, the frequency at which the two instruments measure particle fluxes is not the same (every 2s for the EPT and every 11s for MagEIS). Data from each instrument are averaged on one hour intervals. Thus, we process new data sets with the same time resolution for each instrument. In turn, each time series can be directly compared to one another. Such averages have been performed for a period of three months, from January to March 2014. This time period was selected because it featured the most intense storms of the year and was before the incident of the EPT that occurred in June 2014 until September 2014 (Pierrard et al., 2020). In order to allow a better quantitative comparison between the observations performed by the two

instruments at different spatial locations, the computed hour-average fluxes are directly plotted on log-log scale scatter plots. Moreover, the outer belt was segmented in narrower 'shells', centered on a given value of L and with a width of dL = 0.5. In the discussion of the next section, the L shells that are considered will be labeled by the center L value of the shell. Although relatively wide, this shell width allows to compensate for the rather small period of time used in this analysis. This ensures that enough points are present in the comparison to keep its statistical significance. It is then possible to perform a linear regression on these new data sets in order to compute the Pearson correlation coefficients between the observations of the two instruments. The equation of the regression line is given by:

$$\log_{10}\left(\bar{\phi}^i_{EPT}\right) = \beta_0 + \beta_1 \log_{10}\left(\bar{\phi}^j_{Mag}\right),\tag{1}$$

where $\bar{\phi}^i_{EPT}$ and $\bar{\phi}^j_{Mag}$ are respectively the hour-averaged differential electron fluxes computed from EPT and MagEIS, $i$ and $j$ denote the energy channel selected for the corresponding instruments, $\beta_0$ is the intercept of the regression line and $\beta_1$ is the slope.

It is also useful to compare the integral flux ($\#/(\text{s cm}^2)$) of electrons retrieved with the two instruments. This can be easily done, given the differential flux. Strictly speaking, we integrate the differential flux with respect to the energy and on all solid angles. In practice, we proceed to the following sum,

$$\phi_{int}(E > E_0) = 4\pi \sum_{i=0}^{N} \phi_{diff}(E_i)\, \Delta E_i\tag{2}$$

where $\phi_{diff}(E_i)$ is the differential flux measured in the energy bin $i$ and $\Delta E_i$ is the width of the channel $i$. Thus, the integral flux does not depend on the energy anymore, although it depends on the lowest energy threshold ($E_0$) taken in the sum given above (this is also a consequence of the decrease of the differential flux with the energy). After having retrieved the integral flux, time averages can be computed in order to compare the two instruments.

## 3    Results and discussion

### 3.1    Analysis of the evolution of EPT and MagEIS observations in 2014

EPT and MagEIS have operated simultaneously during six years, between 2013 and 2019. Both instruments were operational during the year of maximum solar activity, in 2014. Figure 1 shows EPT measurements of energetic electron fluxes in the radiation belts, as a function of time and the McIlwain parameter L, throughout 2014 for two different energy channels, 500-600 keV and 1000-2400 keV on top and middle panel respectively. The bottom panel on the graph shows the evolution in time of the Disturbed Storm Time (Dst) index in 2014. This index characterizes the intensity of the horizontal component of the magnetic field at the surface of the Earth in equatorial regions, and is widely used to measure the intensity of geomagnetic storms. The white area in the EPT fluxes corresponds to a lack of observations from June to September. This "hole" in the

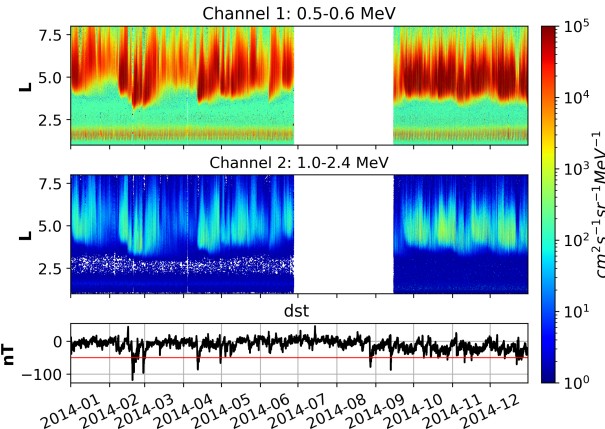

**Figure 1.** EPT electron differential fluxes as a function of time and L throughout 2014, for two different energy channels. Top: channel 1 (0.5-0.6 MeV). Middle: channel 5 (1.0-2.4 MeV). Bottom: Dst index as a function of time where red line corresponds to the constant Dst of -50 nT.

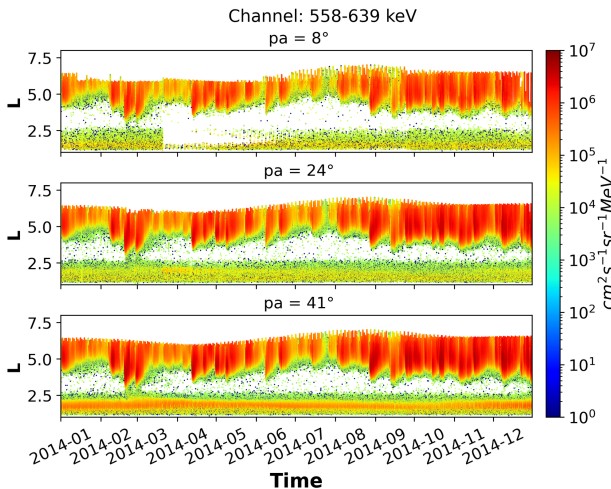

**Figure 2.** MagEIS corrected level 3 pitch angle resolved data as a function of time for 2014, as in the previous figure. The electron flux is measured in a single channel centered at 604 keV for different values of the pitch angle. From top to bottom, each panel shows fluxes measured in increasing pitch angle bins, [0°, 16.36°], [16.36°, 32.72°], [32.72°, 49.09°].

data was caused by an incident on one of the sensors of the EPT. The origin of this problem remains unknown, since no large
storms, nor Solar Energetic Particle (SEP) events were observed at the time.

Because PROBA-V is travelling on a LEO orbit at 820 km, the EPT can only observe inner belt fluxes in the South Atlantic Anomaly (SAA), a region where the geomagnetic field is weaker and trapped particles can penetrate to lower altitudes. Except during extreme events, fluxes in the inner belt are quite steady.

While 2014 was the year of maximum solar activity, it can be seen both in the flux and the Dst temporal variations of Figure 1,
that it was a relatively quiet year in terms of geomagnetic activity. Indeed, only 10 medium storms ($-100$nT $\leq$ Dst $< -50$nT)
were observed and only one intense storm (Dst $< -100$nT) was recorded on February 19. This is not surprising since the
highest frequency of large storms is reached in the declining phase of the solar cycle (Mansilla, 2014; Pierrard et al., 2014).
February was the month featuring the largest geomagnetic storms of the year, the one that occurred on February 19, and another
one, on the 27$^{th}$ during which the Dst index dropped to -96 nT. Both events were caused by Solar Energetic Particle (SEP)
events (https://umbra.nascom.nasa.gov/SEP/). While these storms were responsible for large variations of electron fluxes in
the outer belt, no storms in 2014 was intense enough to inject electrons in the inner belt, where fluxes steadily decrease during
the year, unlike in 2015 (Pierrard et al., 2020). The year 2014 can also be split into two periods characterized by different
geomagnetic activity. During the first period, from January to August, low averaged geomagnetic activity is detected, with a
mean Dst value of $\sim -6,8$ nT. The second period, extending from September to December, features a higher geomagnetic
activity, with a mean Dst value of $\sim -19,3$ nT. However, the storms that took place during this period were less intense.
Because fluxes in the outer electron belts are strongly dependent on the geomagnetic activity, this distinction can also be seen
in the evolution of the flux intensity in Figure 1.

Figure 2 illustrates the RBSP/MagEIS electron differential fluxes observed during 2014 (same year as in Figure 1) for E =
604 keV and increasing pitch angle bins in each panel (from top to bottom [0°, 16.36°], [16.36°, 32.72°], [32.72°, 49.09°] and
will be referred to in the text as pa = 8°, pa = 24°, pa = 41°). This figure shows that the flux variations share similarities with
those observed by EPT. Indeed, electron injections and dropouts occur at the same time, and the location of the inner edge
of the outer belt is the same for observations of both instruments. Despite those similarities between the two data sets, it can
also be seen in Figure 2 that the intensity of the flux observed by MagEIS is higher than with the EPT. In order to precisely
characterize the differences between the observations of the two instruments, a one to one comparison is presented below for
fixed L-shells and energy channels. While fluxes strongly depend on the energy of the electrons, location in the belt and on
the magnetic activity, the minimum flux is always obtained for the lowest value of the pitch angle (Smirnov et al., 2022; Shi
et al., 2016). As illustrated by the different panels of Figure 2, the electron flux in the radiation belts decreases as the pitch
angle of the electrons decreases from pa = 41° to pa = 8°. The decrease in MagEIS electron flux measurements as pitch angle
decreases was shown in previous research (Shi et al., 2016; Smirnov et al., 2022) and obtained with Fokker-Plank simulations
of the L-shell, energy and pitch angle structure of Earth's electron radiation belts during quiet times in Ripoll et al. (2019).

## 3.2 Comparison with AE8 model

Before displaying scatter plots of simultaneous observations from EPT and MagEIS, electron flux measurements from the EPT
are compared to the AE-8 NASA model (Vette, 1991). This is an empirical model of the radiation belts based on averaged
observations from the 60s to the 70s that allows the distinction between periods of minimum and maximum of solar activity.

Figure 3 displays in the top left panel the integral electron fluxes ($> 0.5$ MeV) on the world map as predicted by the AE8
model at an altitude of 820 km and during maximum solar activity. The top right panel in this figure shows the integral flux
of electrons ($>0.5$ MeV, computed with equation 2, see section 2.3) measured by the EPT during February 2014 and averaged

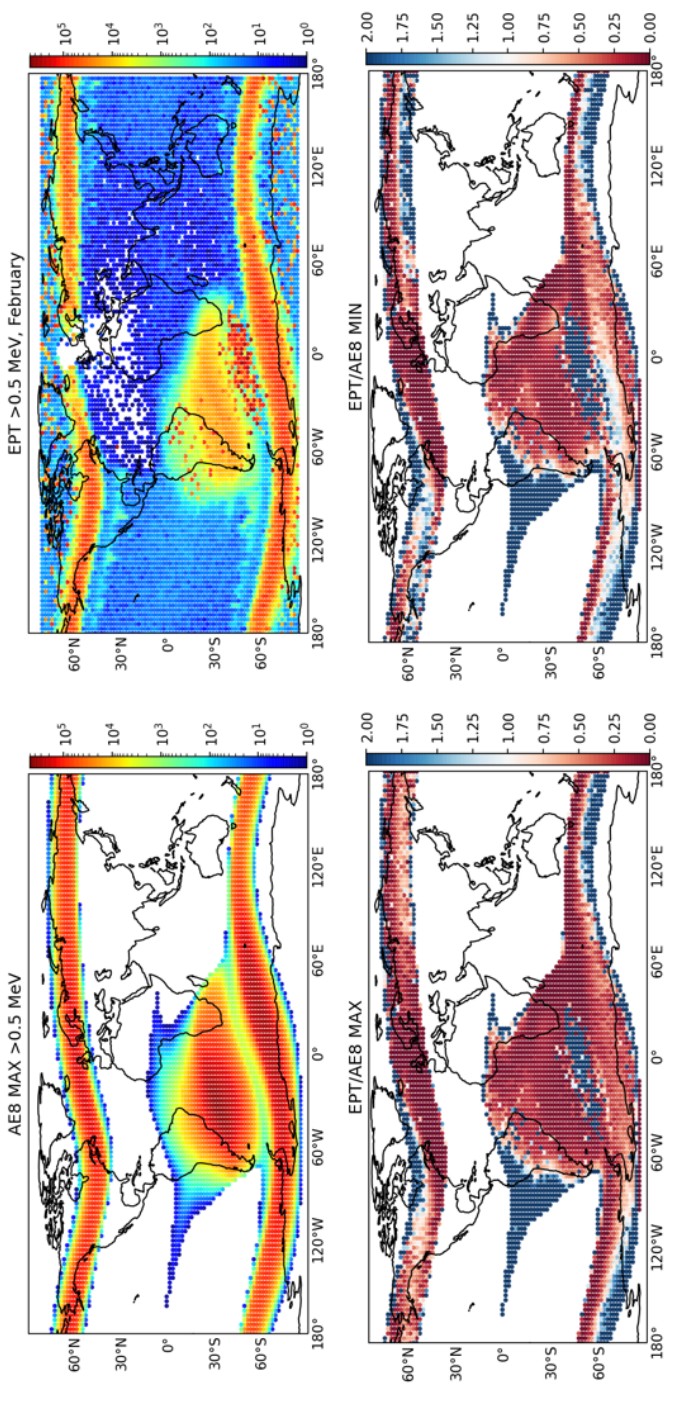

**Figure 3.** Top left: Electron integral fluxes in ($\#/(\mathrm{s\ cm}^2)$) predicted by the AE-8 model at 820 km of altitude during solar maximum. Top right: Integral electron fluxes retrieved from EPT measurements (see equation 2) during February 2014 and averaged on the longitude and latitude in bins ($3°$, $2°$) to match the model resolution. Bottom left: ratio between EPT and AE-8 (solar maximum). Bottom right: ratio between EPT and AE-8 (solar minimum).

on longitude-latitude bins (3° × 2°) corresponding to the resolution of the model. The model is able to reproduce the SAA and the polar horns at high latitudes. Those regions correspond to the penetration of the outer radiation belt at low altitudes. However, the AE8 model does not show the reduced fluxes in the northern hemisphere caused by the counterpart of the SAA that can be observed with the EPT. There is also a region between the SAA and the southern horn where high intensity fluxes are observed by the EPT. Those points are not representative of the mean flux in the bin throughout February, as they are due to measurements performed during the storms and should not be directly compared with the AE8 model which is incapable to reproduce storm fluxes. Similar points can be observed at very high latitudes. The shape of the SAA predicted by the model is not exactly the same as it is observed by the EPT. Eventhough the "heart" (i.e. the regions of the SAA where fluxes are higher than (10 electrons/(s cm$^2$)) ) is similar in the measurements and in the model, the "arm" of the SAA (i.e the region of the SAA of low flux near the equator between 90°W and 170°W) predicted by the model extending over the Pacific ocean is not seen in the measured data. The same structure extending over Africa is also only seen in the model.

The average of the EPT observations on bins similar to those of the model allows a direct comparison between them. Such a comparison is shown on the bottom panels of Figure 3. These two graphs show the ratio between the observations of the EPT and the fluxes predicted by the AE8 model, both during maximum (left) and minimum (right) solar activity. Note that for this comparison, EPT observations remain the same while only the solar activity in the model is changed. Also, in Figure 3, only the fluxes predicted by AE8 during solar maximum are displayed on the top right panel. Predictions of the model during solar minimum are not shown, since the general structure of the map is conserved while flux intensity slightly decreases in the outer belt and slightly increases in the inner belt. In general, electron fluxes predicted by the model in the SAA and in the horns (red regions) are higher than observed by the EPT, especially for AE8 with maximum of solar activity. Fluxes in the outer belt measured by the EPT are closer to the prediction of the model for minimum of solar activity (lower right panel of Figure 3). However, fluxes measured by the EPT are higher than predicted in the most western part of the SAA (blue region). The position of the observed SAA fluxes does not overlap perfectly with the one of the AE8 model. This is a manifestation of the motion of the SAA (3° per year) in the westward direction as a consequence of the secular motion of the geomagnetic field (Pierrard et al., 2014). Even if this motion is taken into account in the model for which the date has to be specified, it seems that there remains some gap. Higher fluxes measured by the EPT are also seen in the outer edges of the polar horns at various latitudes. This is also due to the fact that the simulated and measured fluxes in the horns do not perfectly overlap in these regions. This means that the fluxes are observed to be higher at high L values and thus at high latitudes than what is predicted by the model. When considering the model for maximum solar activity, more intense fluxes are observed inside the horns. The global overestimation of the model during maximum activity can be attributed to the fact that the amplitude of the 24th solar cycle is much smaller than the precedent ones, which were used to develop the model.

### 3.3 Comparison of outer belt fluxes

Figure 4 shows scatterplot comparison between the differential fluxes of the EPT and MagEIS as obtained with the methodology described in section 2.3. Here only two different energy ranges for electrons are displayed, 500-600 keV for the EPT, 558-639 keV for MagEIS and 1000-2400 keV for the EPT and 970-1279 keV for MagEIS. The channels selected for both instruments

are displayed on each panel of the figure. Each row on this figure also corresponds to a different location in the outer radiation belt given by the L range. In addition, on each panel, two sets of dots are represented, corresponding to different data types from MagEIS. Blue dots are computed with MagEIS level 2 spin-averaged data, not taking electron pitch angle into account, while black dots are computed with MagEIs level 3 pitch angle resolved data, for the lowest possible pitch angle bin, pa = $8°$.

From this figure, the evolution of the distribution of points with respect to electron energy and L values can be studied. First, the alignment of the data is reasonably good and the Pearson correlation coefficients range between 0.79 and 0.9. Moreover, fluxes of electrons decrease with increasing energy, for both instruments, independently of the pitch angle value and the position in the outer belt. However the distribution is shifted downward and to the left. The decrease is not the same for the EPT and MagEIS, as indicated by the rapid decrease of the intercept value ($\beta_0$) of the regression line with energy. While for MagEIS the difference in flux between $558-639$ keV and $970-1279$ keV is about one order of magnitude, the difference is about 3 orders of magnitude for the EPT. In addition, the slope of the regression line ($\beta_1$) is always lower than one, indicating that the variation of the flux intensity is in general larger near the equatorial geomagnetic plane than at all low altitude spanned by PROBA-V, and again independently of the pitch angle and the position in the outer belt. MagEIS measurements are systematically higher than those of EPT, except once for the energy of 500 keV, and at L = 4, only for low fluxes (panel e). Those points correspond to the beginning of January 2014, during which fluxes of electrons were unusually low at this location of the belt.

Figure 4 also shows the evolution of the flux-flux distribution as a function of L. For spin-averaged MagEIS data (blue dots), the variations scale of the flux is much larger at L = 4 than for the higher L values. This is related to the very low fluxes observed in January and the high fluxes associated with the storms of February in this region, leading to a very wide flux range. Such low fluxes were not observed at high L values and are hence not seen in the flux-flux distribution. At L = 5 and L = 6, the distribution of points is very different from the one near the inner edge of the belt. This illustrates the different evolution of electron fluxes in the different regions of the outer belt. Indeed, near the inner boundary, fluxes are relatively low until injections lead to sharp flux increase, whereas higher in the outer belt, electrons fluxes remain more intense even during quieter periods. In addition, at high L values, the figure shows the emergence of vertical structures, for which MagEIS fluxes remain relatively constant while a very sharp decrease is observed for the EPT. These structures are caused by dropout events, which are very rapid depletion of electrons in the outer belt during geomagnetic storms. Such events were extensively studied by Pierrard et al. (2020). Dropout events are thus more intense at low altitude than near the equator. Note that this behaviour can be partly explained by the difference in adiabatic losses of electrons at low altitudes and near the equator. Indeed, during a geomagnetic storm, due to the conservation of the second adiabatic invariant of the motion of trapped particles, the altitude of the mirror points will increase (Tu and Li, 2011). This means that low altitude measurements, such as the ones of the EPT (at 820 km) are affected by such effect, while at the equator, the location of the mirror points does not affect the electron flux. Moreover, as they are more frequent at high L values, the structure related to such events are much more prominent for the two top panels of the figure.

While the pitch angle does not affect the variation of the flux with the energy of electrons, the difference in flux intensity between the two instruments is reduced as low pitch angle values are considered (black dots). The differences of flux intensity between MagEIS and EPT are given in Table 2. Eventough the difference in intensity between the two instruments is reduced

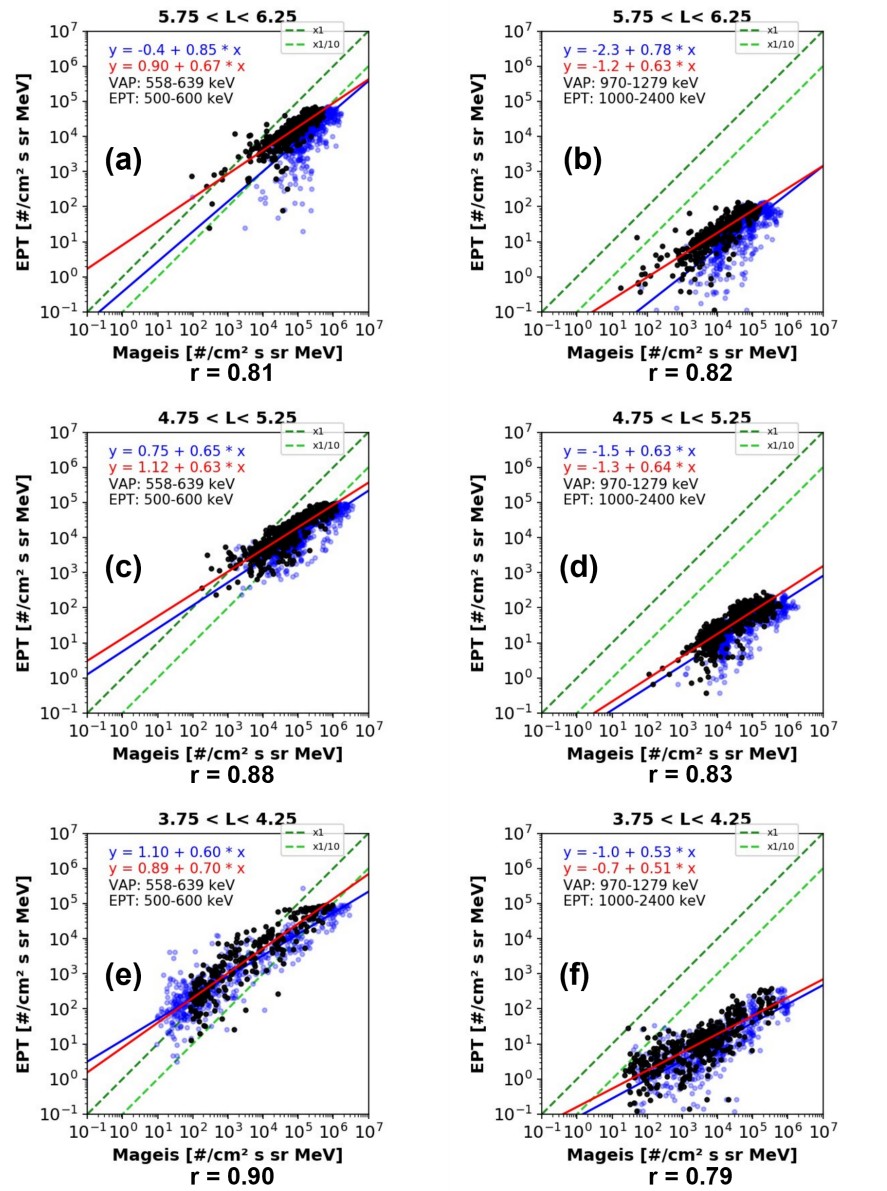

**Figure 4.** Scatterplot of the logarithm of the hour averaged differential electron fluxes from PROBA-V/EPT versus RBSP-B/MagEIS (blue dots for level 2 data and black dots for level 3 data (pitch angle of 8°)) for two different energy channels (column 1: 500 keV, column2: 1 MeV) and locations in the radiation belts (row 1: L = 6, row 2: L = 5, row 3: L = 4). Blue and red lines represent the best fit of the level 2 data and low pitch angle (pa = 8°), respectively. The green lines show perfect linear correlation with a factor of ×1 and ×10⁻¹. Data represented in this graph are from January to March 2014. Pearson correlation coefficient below each panel are computed with low pitch angle values (i.e, black dot distributions).

**Table 2.** This table contains the mean scaling factor (m) between EPT and MagEIS differential fluxes for level 2 data and level 3 for a pitch angle of 8°, such that: $\bar{\phi}_{Mag} = m \times \bar{\phi}_{EPT}$

| Level 2 | 500 keV | 1000 keV | Level 3 (pa=8°) | 500 keV | 1000 keV |
|---------|---------|----------|-----------------|---------|----------|
| L = 4   | 12      | 2206     | L = 5           | 5       | 901      |
| L = 5   | 17      | 3042     | L = 5           | 7       | 1195     |
| L = 6   | 14      | 2263     | L = 6           | 5       | 736      |

when taking low pitch angle equatorial electrons, MagEIS fluxes remain about $10^3$ times higher than those of the EPT at 1000 keV (exact values in the table). In Ginisty et al. (2023b), comparing integral fluxes of relativistic electrons (> 1.6 MeV) from CARMEN 2 -3 at LEO with MagEIS level 2 data show a better agreement in the flux intensity (see Figure 1 and Figure 4 in this reference). CARMEN measures electrons fluxes with an energy > 1600 keV, which corresponds to the energy where the difference in flux intensity between MagEIS and EPT is the largest. However, Figure 4 of Ginisty et al. (2023b), shows that when the flux intensity decreases abruptly, fluxes at low altitude measured by CARMEN reach lower values than MagEIS fluxes, suggesting that sudden decrease of electrons in the outer belt are more important at low altitudes in the outer belt. The LEO orbit of Jason2 and 3 that is located at higher altitude than PROBA-V (1336 km) and with a lower inclination (66°) can at least partially explain the fluxes higher than those of EPT. Indeed, PROBA-V is located at the extreme borders of the radiation belts where the fluxes are lowest and fading quickly away, where fluxes have high gradients. As noted in Pierrard et al. (2021), the trajectory of the particles trapped in the terrestrial magnetic field leads to electron fluxes larger when measured at higher altitudes and at lower latitudes.

Note that at L = 4 for 500 keV, the lowest fluxes are lost for the low pitch angle value. This is due to the fact that for low pitch angle and corrected MagEIS data, a larger amount of data is lost (see Figure 2). It is clear from graph (a) and (b) of Figure 2 that fluxes of electrons with a pitch angle of 8° measured at the equator are more susceptible to the smallest dropouts that occur in the outermost region of the outer belt and are in better agreement with the observations performed at an altitude of 820 km. Indeed, during the month of March, the dropouts that were not observed in the hour-averaged flux computed from spin-averaged data of MagEIS are now observed for low pitch angle electron flux. This leads to much less vertical structures on the scatter plot at low L. In the region of the belt close to the outer edge of the outer belt (L = 6), a substantial diminution of the slope of the regression line can be observed when taking low pitch angle fluxes rather than spin-averaged ones. This decrease is due to the reduction of the number of points corresponding to less intense or non-observed dropouts by MagEIS compared to the measurements performed by the EPT. Because the lower regions of the outer belt are less impacted by the selection of low pitch angle values, such a variation of the slope does not appear at L = 4 and at L = 5.

Because the integral flux is no longer dependent on the energy of the electrons, the comparison of the integral flux computed from EPT and MagEIS measurement is only performed for different values of the McIlwain parameter in the outer belt. A similar analysis to that shown in Figure 4 was carried out for integral fluxes in Winant (2022) but is not displayed here. The results of this comparison are in agreement with the results obtained with the differential fluxes, which should not be surprising,

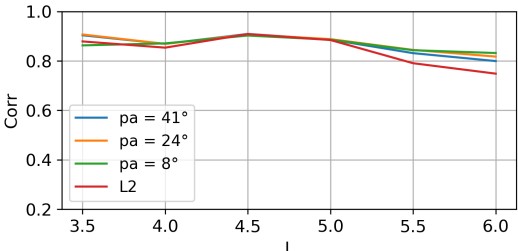

**Figure 5.** Evolution of the correlation coefficients between the logarithm of the integral fluxes computed with the EPT and MagEIS as a function of L and for different values of the pitch angle and for level 2 spin-averaged MagEIS data. The data in this graph have been taken between January and March 2014.

as the integral flux is computed from the differential fluxes. The first observation is that the integral flux measured near the equator is almost always higher than that observed at low altitude, as expected from the bounce motion of the particles along the drift shells (Pierrard et al., 2021). EPT fluxes are higher than those recorded by MagEIS only near the inner edge of the outer belt (L = 4), when both fluxes are relatively low. This is the case for both spin-averaged and low pitch angle electron fluxes. Also, the difference in flux intensity between the two instruments is reduced by considering fluxes of electrons with a pitch angle of $8°$. Indeed, at L = 4, 5, while MagEIS spin-averaged integral flux is respectively 46, 48 times higher than the integral flux obtained with the EPT respectively, small pitch angle fluxes are 16, 18 times higher respectively. The same is true at L = 6 where the spin-averaged MagEIS flux is 32 times larger than for the EPT and becomes 10 times larger when the integral flux is computed with $8°$ pitch angle electrons. This also shows that in the outer part of the outer belt, the difference in flux intensity between MagEIS and the EPT is smaller than for the center and the inner part of the belt. This is valid for both spin-averaged and pitch angle resolved data. A comparison of measurements of the ICARE-NG detector at low (CARMEN 3) and high altitude near the equator (CARMEN 4) showed that the flux intensity at high altitude was about 12 times higher than at LEO, for $L^* = 3.5 - 4.8$ (Ginisty et al., 2023a). For this range of L shells in the outer belt (L = 4 − 5.5), we find a larger difference in the integral flux with the EPT, even for low equatorial pitch angles except at L = 6 (see Table 3). However, in Ginisty et al. (2023a), electron pitch angle was not taken into account. There is thus a relatively large difference with our results. Moreover, we retrieve the integral flux of electrons with an energy > 500 keV while in the case of CARMEN the energy threshold of the integral flux is 1600 keV. As it was mentioned in the differential flux comparison, in the case of the EPT, the flux intensity difference with MagEIS increases with energy. So with the EPT and for energy > 1600 keV, the difference in integral flux with MagEIS will be higher than the results in Table 3.

As it was previously observed, the impact of the selection of low pitch angle electron fluxes is more important in the outer regions of the outer belt (L ≥ 5). An improvement of the correlation is seen compared to the one computed with spin-averaged data, especially at L = 6. Also, comparing small pitch angle fluxes with EPT observations at L ≤ 4.5 leads to a very small decrease in the correlation. The evolution of the correlation between the integral flux computed with MagEIS and the EPT as

**Table 3.** Same as Table 2 but for the integral flux (> 500 keV)

|  | L = 4 | L = 5 | L = 6 |
|---|---|---|---|
| Level 2 | 46 | 48 | 32 |
| Level 3 (pa=8°) | 16 | 18 | 10 |

a function of L is presented in Figure 5. The correlation is computed for spin-averaged data as well as for different pitch angle values, namely pa $= 8°$, pa $= 24°$, pa $= 41°$. This graph shows that even when considering the level-2 spin averaged data from MagEIS, fluxes at low altitude and near geomagnetic equator have a good correlation (corr $> 0.7$) at all L values. This result is in agreement with the results of the comparison between CARMEN and RBSP (Ginisty et al., 2023b), for which the correlation is higher near the inner edge and the center of the outer belt, with a slight decrease near the outer edge. It appears on this figure that for $L > 5$, even by considering electrons with pitch angle of $41°$, the correlation between the instruments is significantly improved. Moreover, by considering successively smaller values of the pitch angle, correlation is further increased. For the lowest pitch angle value, the correlation between the EPT and MagEIS is larger than 0.8 throughout the outer belt. Note that the slight decrease of the correlation at $L = 4$ with decreasing pitch angle is most likely caused by the diminution of the number of points used for the regression with the decrease of the pitch angle. This can clearly be seen in Figure 2. The results obtained here are comparable to the results of the comparison of the measurements from instruments in RBSPs and Arase, which have a similar orbit (Szabó-Roberts et al., 2021).

### 3.4 Conjunction in the inner belt

Finally, the electron fluxes measured by RBSP-A/MagEIS and EPT during the whole period of conjoint operation, i.e. 2013-2019, were employed to compare the fluxes when the satellites were located as close as possible. For this analysis as the EPT data time resolution is 2 seconds and for MagEIS it is 11 seconds, both series of data were averaged to 15 seconds. In order to find the closest space-time conjunctions between both satellites for a better validation, the following conditions were simultaneously imposed between both time series : DL $\leq 0.02$ and DB $\leq 0.01$, where DL and DB accounts for the absolute difference between the corresponding McIlwain L-shell coordinates and Magnetic Fields of the satellites at a particular time. Due to the very different orbits of both satellites, polar at LEO for PROBA-V versus a highly elliptic LEO-MEO for RBSP, after application of the conjunction condition, only some hundreds of observations remain useful to perform the correlation. All are located close to the equator and at very low L ($L \leq 1.4$), as illustrated in Figure 6, inside and outside the SAA.

Figure 7 displays the correlations between the two first energy channels of Table 1. The linear regression (yellow line) demonstrates a relatively good agreement, in particular for the lower energies (500 keV), in line with previous comparisons. The red line corresponds to perfect linear correlation with a factor of 1. The correlation coefficient (indicated at the top of the panels after the linear fit) should be taken with care since the resulted conjunction points are very few (even without the application of any additional flags for MagEIS data), in the region of the South Atlantic Anomaly where contamination from energetic protons can be high, thus imposing corrections for MagEIS measurements (Claudepierre et al., 2015). One can note

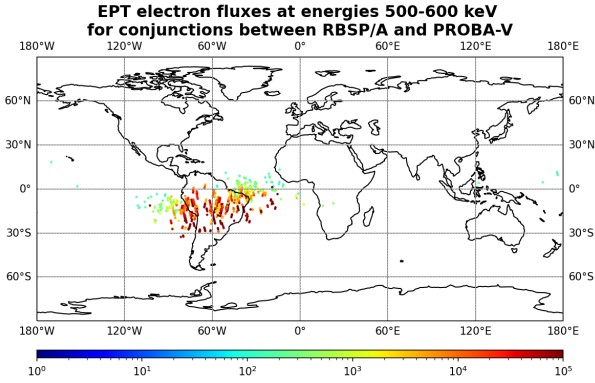

**Figure 6.** EPT electron differential fluxes [#/cm$^2$ s sr MeV] that follow the condition DL $\leq 0.02$ and DB $\leq 0.01$ between the L coordinates and the magnetic fields, respectively, of both satellites RBSP/A and PROBA-V. Left: 500-600 keV for the EPT and 558-639 keV for MagEIS. Right: 700-800 keV for EPT and 692-793 keV.

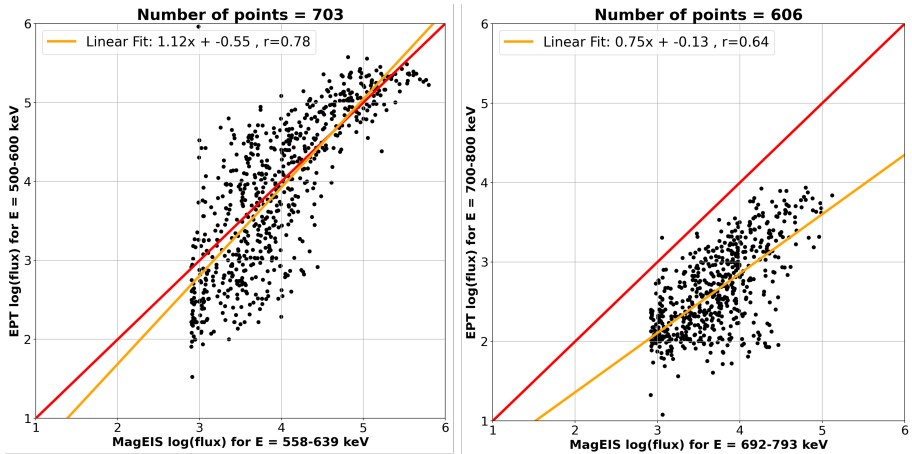

**Figure 7.** EPT electron differential fluxes [#/cm$^2$ s sr MeV] that follow the condition DL $\leq 0.02$ and DB $\leq 0.01$ between the L coordinates and the magnetic fields, respectively, of both satellites RBSP/A and PROBA-V.

that no corrected electron flux lower than $10^3$ $s^{-1}cm^2sr^{-1}MeV^{-1}$ is obtained by MagEIS in the inner belt, while this is not the case for EPT. This can explain why the correlation decreases with the energy since lower fluxes are observed at high energy. In the inner belt, the correction factors between MagEIS and EPT integral fluxes are 0.67 and 11.14 for 500-600 keV and 700-800 keV electrons respectively

## 4   Conclusions

The year 2014 was relatively quiet in terms of magnetic activity compared to the following years. From January to June, geomagnetic activity was low on average, although this period saw the largest storms of the year, especially in February. Conversely, the rest of the year was characterized by a higher magnetic intensity, with lower Dst value on average, but no major event occurred during this period. This can also be seen in the flux intensity measured by the EPT throughout the year, with more intense electron fluxes toward the end of the year. Due to the lack of injections of electrons to very low L values,

the very stable nature of the inner belt is clearly displayed, even for the storm of February 19[th]. However, the variations of electron flux in the outer belt with the geomagnetic activity are well observed for the February storms. In the present work, integral fluxes of electrons obtained from EPT measurements were directly compared with the NASA AE8 empirical model. Because the model can only distinguish between maximum and minimum of solar activity, injections of electrons and protons during magnetic storms and SEP events respectively cannot be reproduced. However, the model is able to well represent the

main features of the radiation belts at low altitudes. Flux intensity in the horns is in general higher in the model than in the observations. This overestimation of the flux by the model is also seen in the SAA. The difference in flux intensity between the model and the observations is much larger in the SAA than in the horns due to lack of injection of electrons in this region in 2014. The comparison of the measurements of energetic electron fluxes in the outer radiation belts was conducted with the use of two science class spectrometers, namely the EPT and MagEIS, on board different spacecraft with very different orbits.

This comparison was performed for various electron energies and locations in the outer belt. Moreover, the effect of the pitch angle for near equatorial electrons was tested between January and March 2014. The comparison between EPT fluxes and spin-averaged fluxes from MagEIS clearly shows that fluxes of electrons decrease with energy, but more importantly, it shows that this decrease is much more abrupt at low altitudes than near the equator. In addition, it is quite evident on the scatter plots that the observations of dropout events are not the same for the two instruments. This difference in measurements is reflected

by vertical structures on the scatter plots, showing sharper decrease of the flux at low altitude. Consideration of low pitch angle (pa = [0°, 16,36°]) electrons has two distinct effects on the results of the comparison. The first one is the reduction of the difference in flux intensity measured by the two instruments at all energy levels and at all L values. Such a reduction in flux intensity is also observed for the integral flux (> 500 keV). Spin-averaged MagEIS fluxes at L = 4, 5, 6 are 46, 48, 32 times higher than EPT fluxes respectively but equatorial low pitch angle fluxes remain one order of magnitude higher than those at

low altitude in the outer belt. At L = 4, 5, 6, MagIES 8° fluxes are 16, 18, 10 times higher than EPT fluxes respectively. This is explained by the motion of the particles along the drift shells: only electrons with low pitch angles are able to reach the low altitudes and high latitude regions where the EPT makes measurements. The second effect is the reduction of the number of vertical structures associated with dropout events, showing that they are more alike than for spin-averaged data. Moreover, even considering spin-averaged data from MagEIS, observations from the two instruments show a good correlation. When

considering low pitch angle electrons, the correlation in the outer region of the outer belt is significantly improved. A relatively good correlation is also obtained in the inner belt in the equatorial plane where the electron fluxes comparisons are performed considering the whole period of mutual operation of both instruments at their closest space-time conjunctions.

The comparison between CARMEN and RBSP performed by Ginisty et al. (2023b) show a better agreement between the integral fluxes intensity measured at low altitude and high altitude than what is found with the EPT, especially considering

that the lower energy threshold of CARMEN fluxes is 1600 keV, the energy at which the difference in intensity between EPT and MagEIS is the largest. The different results obtained in that work and our investigations may partially be explained by the different orbits of the PROBA-V and Jason 2, 3 satellites. Despite those differences, in both studies, dropout events are more important at LEO than at MEO, and a good correlation between LEO and MEO fluxes is found. Finally, for the comparison between EPT and MagEIS, the McIlwain parameter L was used to map the magnetic field lines along which trapped electrons

move. The use of the L parameter instead of L*, used in Ginisty et al. (2023b), or different magnetic external field models for the comparisons with MagEIS data could result in an uncertainty of $L \sim 1$ at high L-shells. These different sources of discrepancy will also be investigated in future works.

*Data availability.* EPT data used in the study are available at https://swe.ssa.esa.int/space-radiation MagEIS data used in this study are available at https://rbsp-ect.newmexicoconsortium.org/data_pub/.

*Author contributions.* AW made the present analyses and wrote the manuscript with the contribution of the other authors. VP conceptualized and supervised the study, and contributed to the interpretation of the results. EB helped in the development of the codes and contributed to the analyses. All authors contributed to writing of the manuscript through reviews and edits.

*Competing interests.* The authors declare that they have no conflict of interest.

*Acknowledgements.* The project 21GRD02 BIOSPHERE has received funding from the European Partnership on Metrology, co-financed

from the European Union's Horizon Europe Research and Innovation Programme and by the Participating States. VP and EB thank the International Space Science Institute (ISSI) and the participants in the ISSI workshop for the project "Radiation belts physics".

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
