# Peer review of "Comparison of radiation belts electron fluxes simultaneously measured with PROBA-V/EPT and RBSP/MagEIS instruments"

_Annales Geophysicae, 2023_

## Referee Comment (RC1)

Dear Editor,

I reviewed the article "Comparison of radiation belts electron fluxes simultaneously measured with PROBA-V/EPT and RBSP/MagEIS instruments" by Alexandre Winant et al. and submitted to AnnGeo.

The article is based on the comparison of electron radiation belt fluxes at low Earth orbit (LEO) and at an elliptic orbit as GTO. Proba-V/EPT is used at LEO. Van Allen Probes are used at GTO (MagEIS instrument). The difference in latitude causes the flux difference that one needs to know in order to better understand the physical processes occurring during the bounce motion of trapped electrons. Fluxes at LEO are also compared with the AE8 model. The comparison of the fluxes is made in a statistical way in the outer belt at both relativistic and ultra-relativistic energies during the year 2014. The work is very well done and results are well established. Another very interesting comparison is done at low L-shells for the few conjunction points that were found. In all cases, differences in flux are well established. Results are at the state of the art and very interesting. The article is also well written but the organization is not fine (see my point 3 below). I am finding missing references too which I would like to see cited, including ones from the authors themselves (my point 1). Also, there are two recent publications, which the authors probably do not know, but with similar results and comparisons made. It is important to relate this study with the two articles. These are the main recommendations I ask the authors to follow (more explanations in following). I will recommend publication of this article once the corrections are made:

1) **Missing references:**

- RBSP: cite Mauk et al. 2013
- MagEIS instrument : Blake et al. 2013
- Proba-V: cite the main article about that mission (is it Cyamukungu et al. 2014 ?)
- Arase: cite Miyoshi et al. 2018 (SSR, ERG mission)
- Radiation belt review: please cite in the introduction and refer to Ripoll, Claudepierre et al. 2020.
- Please refer to Winant master thesis for further information in giving the website link. Is it relevant to do it line 208 where it is written "A similar analysis to that shown in Figure 4 was carried out for integral fluxes but is not displayed here."
- Please refer to Viviane Pierrard, Alexandre Winant, et al. , `Simultaneous Observations of the 23 June 2015` Intense Storm at LEO and GTO Orbits, URSI Radio Science Letters, Vol 4, 2022, doi: 10.46620/22-0016
  In saying that preliminary comparisons were provided and discuss in this article.
- Please acknowledge that the radiation belt often evolves in the plasmasphere and that wave-particle interactions that sculpt the radiation belts will be partly controlled by the local value of the plasmaspheric density. A recent review of plasmasphere modeling is available in Ripoll J-F et al. (2023), Modeling of the cold electron plasma density for radiation belt physics, Front. Astron. Space Sci.
- The decrease in electron flux (MagEIS measurements) as pitch angle decreases is shown in Fig 2 of Ripoll et al. Ripoll, J.-F., Loridan, et al. (2019). Observations and Fokker-Planck simulations of the L-shell, energy, and pitch angle structure of Earth's electron radiation belts during quiet times. Journal of Geophysical Research: Space Physics, 124, 1125-1142. https://doi.org/10.1029/2018JA026111. Please cite.

**2) Similar comparisons have recently been published in EGUsphere and should be referred to and commented**

Comparisons at LEO and GTO have recently been carried in these two articles with links given below. I am assuming is that these two articles are two recent to be known by the authors.

1- Please cite and discuss briefly in the introduction the results in:

https://ieeexplore.ieee.org/abstract/document/10078924

https://www.sciencedirect.com/science/article/pii/S0273117723000029

2- In the main text, in the analysis, please discuss and comment the differences you find between EPT and RBSP with the differences found in the above articles between CARMEN and RBSP. My understanding is that the large differences found this article with EPT-MagEIS are not found between CARMEN (2, 3) and RBSP, in particular at high energy (and knowing differences EPT-RBSP increase with energy increasing). This should be mentioned clearly to the readers and discussed (ideally explained but I am not sure explanations can be given).

**3) The organization of the article is not fine**

- Please consider moving the methodology section before the world map section.
- Why not integrating the world map section within the "results and discussion" section.
- The "Analysis of the EPT observations" is not an adapted title: you work with both EPT and MagEIS….. Why is this section not in the "results and discussion" section.
- I suggest the 2 Instrument section becomes the 2 Instrument and method section, with 2.1 EPT, 2.2 MagEIS, 2.3 methodology
- Then, section 3 of results and discussion can have : 3.1 Analysis of the evolution of EPT and magEIS observations in 2014", 3.2 Comparison with AE8 (formerly "world maps), 3.3 Comparison of outer belt fluxes, 3.4 Conjunction in the inner belt

**4) Writing clearly a main result**

I am asking that in the analysis section of figure 4, a sentence is written and gives the correction factors between MagEIS and EPT for each of the 6 (L-shell, E). That sentence should be also copied in the conclusions. For instance: at L $\sim$ 4, 5, while MagEIS 8° flux integral flux is $\sim$ 20 times higher than the integral flux computed with the EPT. Please make a Table if needed to be clearer.
(About line 255, please give the energy at which the comparison is made.)

So far I read in the conclusions: "but equatorial low pitch angle fluxes remain one order of magnitude higher than those at low 285 altitude in the outer belt.". This is not accurate enough. I want clear numbers according to the (L,E) that is considered.

About the inner belt, it is written "A relatively good correlation is also obtained in the inner belt". Ok, but you could be more precise: the one to one flux correspondence is excellent at 500-600 keV but decreases as energy increases (please give a number of the correction factor).

**5) Other corrections**

- Please rephrase "….the inner belt, mainly composed of energetic protons, extends up to L = 2, depending on the particles energy, and presents a more stable configuration. The outer radiation belt, mainly composed of electrons, is highly sensitive " There is an electron inner belt. Here you oppose a proton inner belt to an electron outer belt.
- All references are weirdly made, for instance: "the detection of a third ultra-relativistic electron belt Baker et al. (2013)". This should be "the detection of a third ultra-relativistic electron belt (Baker et al., 2013)"
- Please rephrase "Like the EPT, the Magnetic Electron Ion Spectrometer (MagEIS) is a science class spectrometer…" EPT is a telescope. MagEIS is a spectrometer. MagEIS uses a magnetic field to deviate and count a given energy range of electrons. This is a fundamentally different instrument….. Please acknowledge. Please read and cite Blake et al. 2013 about MagEIS.
- Define clearly the 'horns", the 'heart' and the 'arms'. You could use examples with (lat, long) if needed.
- Line 105: if we see so well the similarities between fig 1 and 2 why do we need fig 4 and 5. Change of argument: we see some similarities which require a more systematic one-to-one comparison in order to asses precisely the differences. BTW the differences found are large, so don't say here that fluxes agree.
- Legend of fig 3: EPT is not 'computed' even if I understand you use Eq. 2 to build an integral flux. Rephrase.
- It is unclear how you deal with solar min and solar max. Are EPT data selected as such? Or is it just AE8? Please explain? Why AE8 is only shown at solar max and not at solar min? Show it as well.
- When you indicate "L~4,5,6" in many places; you can say first that the center L-shell value of the bin is used at L=4,5,6. This avoids using a '~' which gives the idea the L is not known precisely while it is. Define $L_c$ if you need.
- Don't write ~500 keV. Rather write 500-600 keV. Idem for 1-2.4 MeV. Don't use '~' (everywhere: text and legends)
- Line 226 in: "The same is true at L ~ 6 where MagEIS flux is ~ 30 times larger than for the EPT and becomes ~ 10 times larger when the integral flux is computed with ~ 8° pitch angle electrons". Please write it is the spin-averaged flux.
- The end of line 259 is not understandable because disconnected from the rest of the sentence: "high, and imposed corrections for MagEIS measurements Claudepierre et al. (2015)." It is possible that it is due to the coma ", and" that should be removed.
- Columns inverted in fig 4

[revised manuscript text omitted]

---

## Referee Comment (RC3)

500

[referee-annotated manuscript omitted]

---

## Author Comment (AC3)

**Dear Editor,**

I reviewed the article "Comparison of radiation belts electron fluxes simultaneously measured with PROBA-V/EPT and RBSP/MagEIS instruments" by Alexandre Winant et al. and submitted to AnnGeo.

The article is based on the comparison of electron radiation belt fluxes at low Earth orbit (LEO) and at an elliptic orbit as GTO. Proba-V/EPT is used at LEO. Van Allen Probes are used at GTO (MagEIS instrument). The difference in latitude causes the flux difference that one needs to know in order to better understand the physical processes occurring during the bounce motion of trapped electrons. Fluxes at LEO are also compared with the AE8 model. The comparison of the fluxes is made in a statistical way in the outer belt at both relativistic and ultra-relativistic energies during the year 2014. The work is very well done and results are well established. Another very interesting comparison is done at low L-shells for the fewconjunction points that were found. In all cases, differences in flux are well established.

Results are at the state of the art and very interesting. The article is also well written but theorganization is not fine (see my point 3 below). I am finding missing references too which I would like to see cited, including ones from the authors themselves (my point 1). Also, thereare two recent publications, which the authors probably do not know, but with similar results and comparisons made. It is important to relate this study with the two articles.

These are the main recommendations I ask the authors to follow (more explanations infollowing). I will recommend publication of this article once the corrections are made:

Thank you for your review, comments and suggestions in order to improve our paper. The corrections in the text are in red for the removed sentences and in blue for the added sentences. Our answers are given below. Please note that the line numbers correspond to the lines of the track-changed version of the paper.

**1) Missing references**

Thank you for pointing out missing references. All your suggestions have be added.

- RBSP: cite Mauk et al. 2013 (The reference was added: line 36)
- MagEIS instrument : Blake et al. 2013 (This reference was added: line 91)
- Proba-V: cite the main article about that mission (is it Cyamukungu et al. 2014 ?) (A reference describing the main PROBA-V mission was added: line 37)
- Arase: cite Miyoshi et al. 2018 (SSR, ERG mission) (This reference was added: line 39)
- Radiation belt review: please cite in the introduction and refer to Ripoll, Claudepierreet al. 2020. (This reference was added: line 25)
- Please refer to Winant master thesis for further information in giving the website link. Is it relevant to do it line 208 where it is written "A similar analysis to that shownin Figure 4 was carried out for integral fluxes but is not displayed here." (Added in line 308)
- Please refer to Viviane Pierrard, Alexandre Winant, et al. ,Simultaneous Observationsof the 23 June 2015Intense Storm at LEO and GTO Orbits, URSI Radio Science Letters, Vol 4, 2022, doi: 10.46620/22-0016 (This reference was added: line 64)In saying that preliminary comparisons were provided and discuss in this article.
- Please acknowledge that the radiation belt often evolves in the plasmasphere and that waveparticle interactions that sculpt the radiation belts will be partly controlledby the local value of the plasmaspheric density. A recent review of plasmasphere modeling is available in Ripoll J-F et al. (2023), Modeling of the cold electron plasma density for radiation belt physics, Front. Astron. Space Sci. (This reference was added: line 30)
- The decrease in electron flux (MagEIS measurements) as pitch angle decreases is shown in Fig 2 of Ripoll et al. Ripoll, J.-F., Loridan, et al. (2019). Observations and Fokker-Planck simulations

of the L-shell, energy, and pitch angle structure of Earth'selectron radiation belts during quiet times. Journal of Geophysical Research: Space Physics, 124, 1125-1142. https://doi.org/10.1029/2018JA026111. Please cite.(This reference was added: line 177)

2) Similar comparison have recently been published in EGUsphere and should be referred to and commented

Comparisons at LEO and GTO have recently been carried in these two articles with links given below. I am assuming is that these two articles are two recent to be known by the authors.

Indeed we were not aware of those recent publications during the redaction of this paper. We have read those new studies, added and compared our results with the ones obtained with CARMEN.

1- Please cite and discuss briefly in the introduction the results in:

**https://ieeexplore.ieee.org/abstract/document/10078924**

https://www.sciencedirect.com/science/article/pii/S0273117723000029 (Citations added in the introduction:\_In addition, recent studies have compared electron fluxes observed in the outer radiation belt at low and high latitudes. (Ginisty et al., 2023a) have taken advantage of the Electric Orbit Raising (EOR) of CARMEN4 to geostationary orbit to compare simultaneous observations at LEO of CARMEN3. Both missions where developed by the Centre National d'Etudes Spatiales (CNES) and are fitted with the same instrument, the ICARE-NG detector (Boscher et al., 2014). In this study, a linear relationship between logarithmic values of the electron fluxes  $\geq 1.6$  MeV45 at low and high altitude was found between L\* = 3.5 - 4.8, where L\* is the Roederer parameter (Roederer and Lejosne, 2018). In (Ginisty et al., 2023b) a similar comparison is undertaken between CARMEN2-3 at LEO and RBSP in the outer belt for relativistic electrons ( $\geq 1.6$  MeV). In this work, they report that flux levels are quite similar for both mission, with a good linear correlation between L\* = 3.5 - 4.8]

2- In the main text, in the analysis, please discuss and comment the differences you findbetween EPT and RBSP with the differences found in the above articles between CARMEN and RBSP. My understanding is that the large differences found this article with EPT-MagEIS are not found between CARMEN (2, 3) and RBSP, in particular at high energy (and knowing differences EPT-RBSP increase with energy increasing).

This should be mentioned clearly to the readers and discussed (ideally explained but Iam not sure explanations can be given).

(line 287: comparing our results with CARMEN/RBSP, especially at high energy, since we find a large difference in flux intensity, which is not observed with CARMEN. However, we seen in CARMEN/RBSP comparison that the flux at low altitude decrease more than at high altitudes, which is in agreement with what we have found with EPT)

(line 321: About integral flux comparison, we note the we have much larger correction factor than it was found with the 2 CARMEN missions between low and high altitudes. While for low equatorial pitch angle we the correction factor is similar to what was found with CARMEN, it is not the case with spin-averaged data. Especially at L = 4, 5. Moreover, we have an integral flux with energies > 500 keV while CARMEN measure the integral flux > 1600 keV. Above that energy, the difference that we would find between EPT and MagEIS

would be much higher than what we have computed here.)

(line 336: Despite the differences in flux intensity that we observe with the EPT, throughout the outer belt we find a good linear correlation between low and high altitude measurements, which is in agreement with the results obtained with CARMEN)

**3) The organization of the article is not fine**

Indeed the structure of the paper might have been confusing. The global organization of the paper was modified as you requested.

- Please consider moving the methodology section before the world map section. (The methodology section was moved before the world map section)
- Why not integrating the world map section within the "results and discussion" section. (World map have been moved to the results and discussion section)
- The "Analysis of the EPT observations" is not an adapted title: you work with bothEPT and MagEIS..... Why is this section not in the "results and discussion" section. (The name of this section was changed and moved to the results and discussion section)
- I suggest the 2 Instrument section becomes the 2 Instrument and method section, with 2.1 EPT, 2.2 MagEIS, 2.3 methodology. (The suggested structure is now used in the paper)
- Then, section 3 of results and discussion can have : 3.1 Analysis of the evolution of EPT and magEIS observations in 2014", 3.2 Comparison with AE8 (formerly "world maps), 3.3 Comparison of outer belt fluxes, 3.4 Conjunction in the inner belt (Same as above, the suggested structure was added to the paper)

**4) Writing clearly the main result**

I am asking that in the analysis section of figure 4, a sentence is written and gives the correction factors between MagEIS and EPT for each of the 6 (L-shell, E). That sentenceshould be also copied in the conclusions. For instance: at L~4, 5, while MagEIS 8° flux integral flux is~20 times higher than the integral flux computed with the EPT. Please make a Table if needed to be clearer.

(Tables containing the correction factors were added both for differential fluxes and integral fluxes, for all L, E and pitch angle. For the differential fluxes, a full sentence might be to confusing to read so the reader is directed to the corresponding table.

For the integral flux, the sentence summarizing the correction factor between EPT and MagEIS was changed to explicitly tell the results found in the corresponding table:

Line 315: Indeed at L = 4, 5, while MagEIS spin-averaged integral flux is 46, 48 times higher than the integral flux computed with the EPT respectively, small pitch angle fluxes are 16, 18 times higher respectively. The same is true at L = 6 where the spin-averaged MagEIS flux is 32 times larger than for the EPT and becomes 10 times larger when the integral flux is computed with 8° pitch angle electrons )

**(About line 255, please give the energy at which the comparison is made.)**

(The energy at which the comparison is made was added: line 367)

So far I read in the conclusions: "but equatorial low pitch angle fluxes remain one orderof magnitude higher than those at low 285 altitude in the outer belt.". This is not accurate enough. I want clear numbers according to the (L,E) that is considered.

(line 386: This sentence was modified to be more precise: Such a reduction in flux intensity is also observed for the integral flux (> 500 keV). Spin-averaged MagEIS fluxes at L = 4, 5, 6 are 46, 48, 32 times higher than EPT fluxes respectively but equatorial low pitch angle fluxes remain one order of magnitude higher than those at low altitude in the outer belt. At L = 4, 5, 6, MagIES 8° fluxes are 16, 18, 10 times higher than EPT fluxes respectively.)

About the inner belt, it is written "A relatively good correlation is also obtained in the inner belt". Ok, but you could be more precise: the one to one flux correspondence is excellent at 500-600 keV but decreases as energy increases (please give a number of the correction factor). (The correction factors were added in the text)

**5) Other corrections**

- -Please rephrase ".... the inner belt, mainly composed of energetic protons, extends up to L = 2, depending on the particles energy, and presents a more stable configuration. The outer radiationbelt, mainly composed of electrons, is highly sensitive " There is an electron inner belt. Hereyou oppose a proton inner belt to an electron outer belt. (line 20: True, with this sentence it seem no electrons are found in the inner belt. The sentence was changed to: ...the inner belt, composed of both protons and electrons of high energy, extends up to L = 2, depending on the particles energy,...)
- -All references are weirdly made, for instance: "the detection of a third ultra-relativistic electron belt Baker et al. (2013)". This should be "the detection of a third ultra-relativistic electron belt (Baker et al., 2013)" (The issue regarding the way citations were compiled by Overleaf was fixed)
- -Please rephrase "Like the EPT, the Magnetic Electron Ion Spectrometer (MagEIS) is a science class spectrometer..." EPT is a telescope. MagEIS is a spectrometer. MagEIS uses a magnetic field to deviate and count a given energy range of electrons. This is afundamentally different instrument..... Please acknowledge. Please read and cite Blake et al. 2013 about MagEIS.(line 90: A new line was added to highlight the different principle on which MagEIS is based compared to the EPT i.e. Unlike the EPT, MagEIS relies on uniform magnetic fields to focus electrons and sort their energy on a linear strip of detectors (Blake et al. 2013))
- -Define clearly the 'horns", the 'heart' and the 'arms'. You could use examples with (lat, long) if needed. (explanation added at line 186 for the horns and 193 for the heart and arms).
- -Line 105: if we see so well the similarities between fig 1 and 2 why do we need fig 4 and 5. Change of argument: we see some similarities which require a more systematicone-to-one comparison in order to asses precisely the differences. BTW the differences found are large, so don't say here that fluxes agree.
- (The formulation of the sentence was changes to motivate the systematical comparison between the two instruments: line 167)

In addition, in the abstract, we removed the sentence saying that the fluxes agree and replaced it by: Despite the difference in flux intensity observed by the two instruments, especially at high

energies, a linear relationship with good correlation was found. The correlation is maximum when low pitch angle electrons near the equator are considered.)

- -Legend of fig 3: EPT is not 'computed' even if I understand you use Eq. 2 to build an integral flux. Rephrase. (In the figure caption, the word 'computed' was replaced by 'retrieved', in addition to emphasize the fact that it is not a directly measured by the EPT, a reference to the equation used to retrieve the integral flux was added)
- -It is unclear how you deal with solar min and solar max. Are EPT data selected as such? Or is it just AE8? Please explain? Why AE8 is only shown at solar max and not at solar min? Show it as well.

(We added explanations at line 200)

(We only vary the solar activity for the model while the observations of the EPT are for February 2014. The comparison with AE8 during minimum of solar activity is there to compare if the flux agreement with EPT is better than for maximum of solar activity.

In the text this was clarified: Note that for this comparison, EPT observations remain the same while only the solar activity in the model is changed. Also, in Figure 3 only the fluxes predicted by AE8 during solar maximum are displayed on the top right panel. Predictions of the model during solar minimum are not shown, since the general structure of the map is conserved while flux intensity slightly decreases in the outer belt and slightly increases in the inner belt.

We feel that the addition of the flux map from the model during solar minimum does not bring a major point in the frame of this small comparison. If you still feel that this is required, we will add a figure)

- -When you indicate "L~4,5,6" in many places; you can say first that the center L-shell value of the bin is used at L=4,5,6. This avoids using a '~' which gives the idea the Lis not known precisely while it is. Define  $L_c$  if you need. (This was changed through the entire paper as suggested. A sentence was added in the method section to clarify that when L = 5 refers to the central value of the bin.)
- -Don't write ~500 keV. Rather write 500-600 keV. Idem for 1-2.4 MeV. Don't use '~' (everywhere: text and legends) (This was also changed throughout the text)
- -Line 226 in: "The same is true at L~6 where MagEIS flux is~30 times larger than for the EPT and becomes~10 times larger when the integral flux is computed with~8° pitch angle electrons". Please write it is the spin-averaged flux.
  (Line 317: we specified the that the first correction factor corresponds to MagEIS spin-averaged fluxes.)
- -The end of line 259 is not understandable because disconnected from the rest of the sentence: "high, and imposed corrections for MagEIS measurements Claudepierre et al. (2015)." It is possible that it is due to the coma ", and" that should be removed.

(Line 358: The sentence was changed to: The correlation coefficient (indicated at the top of the panels after the linear fit) should be taken with care since the resulted conjunction points are very few (even without the application of any additional flags for MagEIS data), in the region of the South Atlantic Anomaly where contamination from energetic protons can be high, thus imposing corrections for MagEIS measurements)

-Columns inverted in fig 4 (columns are now in correct order)

---

## Author Comment (AC5)

This paper has shown a comparison between PROBA-V/EPT and RBSP/MagEIS instruments which had observed different altitudes in the radiation belts. The authors used L-shell sorted data to compare two satellite data, and a result of comparison seems to be good. The paper is well written and organized, but I have a couple of questions which the authors may consider.

We thank you for your review and suggestions to improve our work. Please note that all the line numbers refer to the track-changed version of the paper.

1) Adiabatic effect at the low-altitude satellite observation

Tu and Li[2011, JGR, 10.1029/2011JA016468] has discussed the adiabatic loss effects at the low altitudes through variations of the mirror point altitudes. I suppose that the PROBA-V/EPT data has included such effect which causes differences from the RBSP observations. Could you discuss this point, especailly for the low flux time interval of PROBA-V?

This is a very interesting remark. The data of the EPT (at 820 km) will indeed be affected by the altitude increase of the mirror points. We added the following text to the paper at line 293:

*"Note that this behaviour can be partly explained by the difference in adiabatic losses of electrons at low altitudes and near the equator. Indeed, during a geomagnetic storm, due to the conservation of the second adiabatic invariant of the motion of trapped particles, the altitude of the mirror points will increase (Tu and Li, 2011). This means that low altitude measurements, such as the ones of the EPT (at 820 km) are affected by such effect, while at the equator, the location of the mirror points do not affect the electron flux."*

2) L-shell definition

The authors have used McILwain L value for comparison of both satellites? Is this enough to compare two satellite data at different altitudes? I suggest the authors should use Roeder L* using the time-variable Tsyganeneko-04 or later model and include dicsussion how the authors confirm the accuracy of the field line mapping between two satellites.

Indeed, for both instruments, we used the McIlwain parameter L. The reason for it was because those values are directly provided in both data sets. For MagEIS data, both McIlwain and Roederer parameters are given. However, it is not the case for the EPT, for which only the McIlwain parameter is given. We preferred to make comparisons on similar quantities L than introducing computations of the Roederer L* based on possibly different magnetic field models. Moreover, for L<6 as considered in the present article, no major changes are expected. But the suggested approach is interesting and could be considered in further work. However implementing it in this work would require a lot of efforts for almost no expected improvement.

Minor comments:

page 2: Please include XEP as well as HEP for Arase and relavant references (Miyoshi et al., 2018, , Earthe Planet and Space, doi: 10.1186/s40623-018-0862-0, Mitani et al, 2018, Earthe Planet and Space, doi 10.1186/s40623-018-0853-1, Higashio et al., 2018, Earthe Planet and Space, doi:10.1186/s40623-018-0901-x), and MagEIS (Blake et al., Space Sciece Review, 2013, doi:10.1007/s11214-021-00855-2)

We added those references to our paper, at line: 36, 39, 40, 41